# The Quality of Life of Children with Epilepsy and the Impact of the Disease on the Family Functioning

**DOI:** 10.3390/ijerph19042277

**Published:** 2022-02-17

**Authors:** Anna Rozensztrauch, Aleksandra Kołtuniuk

**Affiliations:** 1Department of Nursing and Obstetrics, Division of Family and Pediatric Nursing, Wroclaw Medical University, Poland 1, 51-618 Wrocław, Poland; 2Department of Nursing and Obstetrics, Division of Internal Medicine Nursing, Wroclaw Medical University, Poland 1, 51-618 Wrocław, Poland; aleksandra.koltuniuk@umw.edu.pl

**Keywords:** quality of life, child, epilepsy, parents, family

## Abstract

Epilepsy is a neurological chronic disease, which negatively affects physical, psychological and social functioning of children and their families. The main objective of this study was to assess the quality of life (QoL) in children with diagnosed epilepsy and the impact of a child’s disease on the functioning of the family. Method: A cross-sectional survey involved a total of 103 legal guardians of children with diagnosed epilepsy. QoL was measured by PedsQL 4.0, with appropriate forms for specific age groups, the impact of a child’s condition on the functioning of the family was measured by PedsQL 2.0 Family Impact Module, and the authors’ own questionnaire was used to collect sociodemographic and medical data. Results: Subjects reported a decreased level of family daily activities (total score: 32.4 out of 100, SD = 26.5) and relationships (total score: 55.63 out of 100, SD = 24.03). QoL in children aged 5–7 years is lower by an average of 11.956 points as compared with children aged 2–4 years. Comorbidities had a significant impact (*p* < 0.05) on QoL in all domains. The overall QoL has reported a low score of 46.42 out of 100, respectively (SD ± 20.95), with the highest mean scores reported for the social functioning (total score: 49.4, SD = 27.3) and the physical functioning (total score: 49.4, SD = 28.4) and with the lowest mean score reported for the work/school functioning (total score: 42.3, SD = 27.8). Conclusions: Child’s epilepsy shows a considerable negative impact on the QoL of children and family functioning.

## 1. Introduction

According to the World Health Organization, approximately 7.60 per 1000 individuals experience epilepsy during their lifetime, with the condition affecting around 70 million people of all ages worldwide [1]. The maximum incidence of epilepsy, i.e., 102 per 100,000 cases annually, occurs in the first year of life, similar to the age range of 1 to 12 years. The incidence of epilepsy in children aged 11–17 is 21–24 per 100,000 cases [2,3]. Epilepsy is the most common neurological brain disorder seen in children. The diagnosis of epilepsy can be challenging as many epilepsy imitators have to be considered. Neuroimaging and electroencephalography seem to be critical in determining the etiology of the condition. In addition, genetic testing is often useful, especially in the case of early-life epilepsies [4]. While one-third of cases of epilepsy are caused by acquired injuries (injury during the labour, brain injury or tumour), the remaining cases are believed to be due to genetic factors, including monogenic and polygenic inheritance [5]. The monogenic forms of epileptic disorders tend to occur earlier in life and cover a broad clinical spectrum, from mild, self-limited epilepsy (epilepsy caused by inborn errors of metabolism) to severe early-onset encephalopathy and epilepsies linked to other neurodevelopmental problems [6,7]. Early diagnosis is crucial to reducing the risk of recurrence, ensuring a better prognosis and optimizing treatment in order to ensure the best possible quality of life for the child and their parents or legal guardians. Furthermore, establishing a diagnosis of epilepsy in early childhood has a profound impact on the child’s psychological and physical development. It may reduce certain limitations in the lives of the child’s parents or legal guardians and improve general well-being and quality of life [5,6]. It has been found that specific limitations associated with a disability or impairment may be an important factor affecting mobility, the choice of profession and family planning decisions [7]. Studies have shown that restrictions and limitations may also be a risk factor for depression in mothers of children with epilepsy [8,9].

Quality of life (QoL) depends on numerous factors. In the case of children with epilepsy, some of those factors are directly linked to frequent seizures and adverse effects of the medication taken [10,11,12,13]. QoL is a complex and multidimensional construct that represents the general well-being of an individual by outlining individual positive and negative aspects of life [14]. At the contemporary advanced stage of the medical care system, development emphasis is placed not only on direct treatment effects but also on the patient’s QoL and, at the same time, on their families’ and legal guardians’ QoL. Importantly, epilepsy retards the achievement of independence in a child and makes social relationships and cognitive processes more difficult. These are factors that influence the child’s individual development, and, therefore, they should be evaluated on a constant basis during the treatment.

There is a need for a deeper analysis of an epileptic child’s QoL as learning their psychosocial functioning should provide a possibility for taking the right actions in terms of care for these children. Studies on the epileptic child’s QoL may help with a potential improvement of the care as well as contribute to the creation of novel models of support and work with the children or elaborate a completely new schedule of support provided for the benefit of these children and their families. The child’s disease is a reason for the worsening of its QoL, which is manifested by a decrease in its mental and physical skills. The use of the correct treatment, its normal course, and, first and foremost, the adequate conditions for a child’s harmonic development may lead to the improvement of the patient’s functioning and reduce the negative effect of the disease on the family’s functioning.

The aim of our study was to assess the quality of life in children with diagnosed epilepsy and the impact of a child’s epilepsy on the functioning of their family.

## 2. Materials and Methods

### 2.1. Population and Study Design

The study was conducted in 2019 at the paediatric department with a sub-department of neurology at one of the leading specialist medical centres in Poland. The study included a total of 103 parents and legal guardians of children with diagnosed epilepsy (Table 1 and Table 2).

The most frequent kind of seizures were represented by partial seizures (63%). At the same time, generalized tonic and clonic seizures constituted only 28% of all these episodes. In just 4% of patients attacks of unconsciousness were reported. Mean age of the disease diagnosis was 3.5 years. Among patients, seizures were divided into three intervals of frequency, i.e., recorded every day, once to five times a month or at least once a year in 3.8%, 19% and 78.8% of the patients, respectively. Only 21.2% of children studied had no seizures for the full year. The majority of children (87%) did not yield any disorders in their psychomotor development evaluated on the basis of child’s health balances by a paediatrician and a neurologist.

The study was conducted by means of a diagnostic survey using standardised questionnaires: a generic questionnaire assessing QoL in children (PedsQL 4.0), with appropriate forms for specific age groups (2–4, 5–7, 8–12, 13–18, 19–25, 25 or over) [15,16,17,18,19], a questionnaire assessing the impact of a child’s chronic health condition on the functioning of their family (PedsQL 2.0 Family Impact Module) [20,21,22,23] and our own questionnaire with questions concerning sociodemographic data. The respondents were selected using purposive sampling. The Polish version of all instruments was used. The PedsQL questionnaire was developed for measuring quality of life in children suffering from acute and chronic disease. The Paediatric Quality of Life Inventory (PedsQL 4.0TM) and The PedsQLTM Family Impact Module have good reliability, validity and sensitivity (internal consistency reliabilities exceeded the minimum alpha coefficient standard of 0.7).

The inclusion criteria for the study were as follows: epilepsy diagnosed in the child in accordance with the ICD-10 criteria, declaration that the respondent is the main caregiver of the child and resides permanently with the child concerned, absence of a diagnosed mental illness in the respondent. The exclusion criteria were as follows: an incomplete questionnaire, lack of written consent to participate in the study. The respondents received the questionnaires in paper form for self-administration and were provided with a complete information sheet stating that participation in the study was voluntary and anonymous.

### 2.2. Measuring Instruments

#### 2.2.1. The Paediatric Quality of Life Inventory (PedsQL™) 4.0 Generic Core Scales

PedsQL 4.0 is used to measure health-related quality of life in children and adolescents, both healthy and those with chronic and acute health conditions, aged between 2 and 18 years. The questionnaire allows for the assessment of the functioning, and hence also deficits, of young patients in four areas: physical functioning, emotional functioning, social functioning and work (preschool/school) functioning. Moreover, the questionnaire yields two summary scores: the psychosocial health summary score, which comprises the average of items in the emotional, social and work/school functioning subscales, and the total QoL score (which comprises the average of items in all four subscales). There are no standards allowing for the interpretation of the subscale scores as low, medium or high. However, scores on each of the subscales range from 0 to 100, which makes it possible to compare them. A higher score on a given subscale represents better functioning in a given domain of life.

#### 2.2.2. PedsQL 2.0 Family Impact Module

The PedsQL Family Impact Module has been developed to assess the functioning of respondents in eight areas: physical functioning, emotional functioning, social functioning, cognitive functioning, communication, worry, daily activities and family relationships. The questionnaire also yields three summary scores: the parent health-related quality of life summary score (which is calculated by averaging items in the physical, emotional, social and cognitive functioning scales), the family functioning summary score (which is calculated by averaging items in the daily activities and family relationships scales) and the total impact score (which is calculated by averaging items in all eight scales).

#### 2.2.3. Authors’ Own Questionnaire

Our own questionnaire comprised 7 questions, including those concerning sociometric data such as: age of the child, age of the parent/legal guardian, gender of the parent legal guardian, education of the parent/legal guardian, place of residence and marital status of the parent/legal guardian and a question concerning clinical data, i.e., about whether the child has any comorbidities.

### 2.3. Statistical Methods

An analysis of quantitative variables (i.e., expressed as numbers) was carried out by calculating means, standard deviations, medians and quartiles as well as minimum and maximum values. Qualitative variables (i.e., not expressed as numbers) were analysed by calculating the number and percentage of occurrences of each value. The values of quantitative variables in two groups were compared using Student’s *t*-test (where the variable concerned had a normal distribution in the groups analysed) or the Mann-Whitney test (otherwise). The values of quantitative variables in three or more groups were compared using an analysis of variance—ANOVA (where the variable concerned had a normal distribution in the groups analysed) or the Kruskal-Wallis test (otherwise). Where statistically significant differences were detected, a post-hoc analysis was carried out (using Fisher’s LSD test where the distribution was normal or Dunn’s test where the distribution was not normal) to determine which groups differ from one another. Pearson’s coefficient was used to analyse correlations between two quantitative variables (where the two variables had a normal distribution). In cases where at least one of the variables did not have a normal distribution, Spearman’s coefficient was used.

The strength of relationships was interpreted as follows [24]:|r| ≥ 0.9—very strong relationship;0.7 ≤ |r| < 0.9—strong relationship;0.5 ≤ |r| < 0.7—moderate relationship;0.3 ≤ |r| < 0.5—weak relationship;|r| < 0.3—very weak relationship (negligible).

A multi-factor analysis of the independent impact of a number of variables on a qualitative variable was carried out using linear regression. The results were reported as the values of the regression model parameters with a 95% confidence interval. The normality of distribution of variables was tested using the Shapiro-Wilk test. A level of significance of 0.05 was used in the analysis. Therefore, all *p*-values of less than 0.05 were interpreted as indicating significant relationships. The analysis was performed using the R software, version 3.4.3.

### 2.4. Ethical Aspects

The study was conducted in accordance with the guidelines of the Declaration of Helsinki and approved by the Institutional Review Board (or Ethics Committee) of the Wroclaw Medical University (No. KB–122/2017). The authors declare no conflict of interest.

## 3. Results

The results of socio-demographic characteristics are summarized in Table 1. The majority of respondents were female (93.20%), with the median age of 35.79, SD: 5.89. Most of them (99%) were educated more than primary school level. Over 80% were in a relationship, and 65.05% lived in an urban area.

The age of respondents diagnosed with epilepsy ranged from 2–25 years old. The majority of them had comorbidities (76%) and lived in an urban area (65.05%). Child’s characteristic is presented in Table 2.

### 3.1. Analysis of the Quality of Life in the Children Studied Based on Standardised Instruments

#### 3.1.1. Quality of Life as Assessed Using the PedsQL^TM^ Generic Core Scales

QoL is a multidimensional construct which encompasses several domains. The physical domain includes symptoms such as lack of energy, pain, participation in physical activities, limitations in activities or self-care. In psychosocial functioning, we evaluate how the child is functioning in social life, in contacts with friends or close and distant family. Functioning in the social environment is also associated with the choice of profession and, consequently, taking up a job. The analysis of the data obtained from the PedsQL questionnaire showed that the mean parent-proxy reported total QoL score of the children was 46 (on 0–110 scale), which indicates that overall quality of life has been significantly reduced, with the highest mean scores reported for the social functioning domain (total score: 49.4, SD = 27.3), the physical functioning domain (total score: 49.4, SD = 28.4) and with the lowest mean score reported for the work/school functioning domain (total score: 42.3, SD = 27.8) (Table 3).

The post-hoc analysis showed that the mean school/preschool functioning score reported for children aged 2–4 years was significantly (*p* = 0.012) higher as compared to that reported for children aged 5–7 years and those aged 8–12 years. The analysis also showed that the mean psychosocial health summary score was significantly (*p* = 0.047) higher for children aged 2–4 years than for children aged 8–12 years (Table 4).

Importantly, we found that there is a significant association between parental age and the child’s work/school functioning (*p* = 0.038, r = −0.0204). The older the respondent, the lower the QoL of their child in the work/school functioning domain (Table 5).

Parental education level, place of residence and marital status did not have a significant impact on family functioning in the analysed domains of the PedsQL questionnaire.

#### 3.1.2. Impact of Epilepsy on Family Functioning as Assessed Using the PedsQL Family Impact Module

The analysis of the data obtained from the PedsQL Family Impact Module questionnaire allowed for assessing the functioning of the respondents in eight domains. Higher scores indicate better family functioning and caregivers’ QoL. It showed that the parents included in the study reported the highest scores for the cognitive functioning domain (total score: 52.8, SD = 27.9) and the family relationships domain (total score: 55.63, SD = 24.03) and reported the lowest scores for the worry domain (total score: 24.1, SD = 19.5) and the daily activities domain (total score: 32.4, SD = 26.5) (Table 6). The medical condition of the children studied has a significant impact on the functioning of their families (Table 7). The mean summary scores were around 40–45 on a 0–100 scale.

### 3.2. Analysis of the Relationship between Quality of Life and the Presence of Comorbidities

Comorbidities had a significant impact (*p* < 0.05) on QoL in the children with epilepsy in all domains (Table 8). Additionally, comparing the mean values, it can be stated that comorbidities affect all domains with score below 60 on 0–100 scale. Domains most affected are work/school functioning (total score 38.78 out of 100, SD = 27.29), emotional functioning (total score: 41.03, SD = 20.99) and psychosocial health (total score: 41.47, SD = 19.5).

### 3.3. Linear Regression

Factors affecting the total QoL score in the PedsQL questionnaire

The linear regression model showed that the following factors are independent predictors of quality of life as measured by the total QoL score (*p* < 0.05):Score on the social functioning scale of the PedsQL—Family Impact Module questionnaire. Each additional point is associated with an increase in QoL by an average of 0.257 points.Score on the cognitive functioning scale of the PedsQL—Family Impact Module questionnaire. Each additional point is associated with an increase in QoL by an average of 0.232 points.Age of the child. QoL in children aged 5–7 years is lower by an average of 11.956 points as compared with children aged 2–4 years.Number of children in the family. Where there are three or four children in the family, QoL is higher by an average of 15.957 points as compared with families with only one child (Table 9).

## 4. Discussion

A chronic disease may be the cause of many changes in all spheres of a child’s life, from daily duties through emotional and cognitive development to self-image and relationships with other people. In particular, it may have a negative effect on his contact with peers or functioning in school or in the family. Epilepsy compromises the quality of life of patients because, as a chronic disease, it affects the patient’s daily life, both personally and socially. The present study was designed in such a way as to allow for the identification of the determinants of quality of life in children with epilepsy and the impact of the condition on family functioning. We assessed selected parameters using age-specific questionnaires. We found that QoL in children with epilepsy was highest in the social and physical domains and lowest in the preschool/school/work functioning domain. Older children had a lower overall QoL, as measured by the total QoL score, as compared with younger children. The mean preschool/school functioning score was higher for children aged 2–4 years than for children aged 5–7 years and those aged 8–12 years. Similar associations were found for the psychosocial health domain. The mean psychosocial health summary score for children aged 2–4 years was much higher as compared with that for children aged 8–12 years. In their study, Nadkarni et al. [25] reported similar findings. According to the authors, the worse QoL in the emotional, social and behavioural domains observed in older children with epilepsy may be due to the fact that older children are more likely to perceive a stronger negative impact of the condition on their lives and find it more difficult to cope with its complications, as they are more aware of seizures and their consequences. Therefore, older children with epilepsy have a more negative attitude towards their condition. In their study, Riechmann et al. [26] found no differences in QoL relative to the age of patients. It has been found that, in children with epilepsy aged over 8 years, quality of life is related to their mental health and peer support and is not associated with the severity of seizures. This finding is consistent with literature reviews by Huebner et al. [27] and Fayed et al. [28], which showed that recurrent positive everyday experiences (e.g., at school or in the family) are more important to satisfaction with life in children than major life stressors (such as the diagnosis of epilepsy or seizures). Lowering life satisfaction caused by a specific limitation in social functioning may additionally result in the patient’s impoverishment, minimizing contacts with friends or extended family. This may result in the phenomenon of loneliness and isolation from life and the lack of acceptance of one’s own illness. Therefore, there is a lot of potential for clinicians to improve or maintain young patients’ positive perceptions of their lives in the context of an epilepsy diagnosis through the provision of psychosocial care [29]. The direct relationship between peer support and QoL suggests that interventions targeted at peers may have an immediate effect on QoL in children with epilepsy. The way the patients react to the disease is very individual. Awareness of the disease can provoke negative feelings, even when the therapeutic effect is good and the seizures are fully controlled. The method of treatment is not indifferent to the quality of life of the patients.

Our study did not show an association between QoL in children with epilepsy and the level of parental education. In contrast, Masri et al. [30] found that the higher the education level of parents of children with epilepsy, the better the parents’ knowledge of the condition. These authors also showed that there is an association between positive parental attitudes and behaviours towards epilepsy and a higher parental education level. Moreover, the better the attitudes of parents, the better they function in everyday life and, consequently, the better the quality of life of their children, who thus have a closer bond with their parents. We found no statistically significant association between QoL in children with epilepsy and their place of residence. In contrast, Nadkarni et al. [25] found that the overall QoL in children with epilepsy living in rural areas was lower as compared to that of children living in urban areas. This may be due to the stigma of epilepsy that is present in rural areas, which worsens the functioning of children with epilepsy living in rural areas as compared to those living in urban areas. In addition, the treatment and control of epilepsy may be difficult in rural families due to insufficient access to centres with a high degree of specialized neurological care.

The analysis of our findings showed that there is an association between QoL in children with epilepsy and the presence of comorbidities. Children with comorbid conditions had a significantly lower QoL in the physical functioning, social functioning, psychosocial health and preschool/school/work functioning domains as well as a significantly lower overall QoL, as measured by the total QoL score, as compared with children with epilepsy only. These findings are consistent with the results of previous meta-analyses, in which the presence of comorbidities was identified as a significant factor reducing QoL [30,31]. What plays an important role in the life of a child with a medical condition is their family, who help them deal with the condition. The family is the foundation of society. It is the smallest basic unit in which we learn to live and function. The family environment is the first educational environment in a child’s life, which shapes their feelings and attitudes. One of the most valuable features of a family would be mutual interactions among its members (in this case parents and children), which build understanding, trust and the feeling of security in the family environment. Each action that is taken and each situation that occurs in a family has an impact on all its members. Strong bonds are formed between family members, including those between parents and children and those between siblings. All those relationships become very much pronounced in the context of difficulties that disturb the peace and harmony in family life. Undoubtedly, one example of such adversity is a child’s illness [32,33].

In the present study, we also analysed the impact of a child’s health condition on family functioning. We found that the parents of children with epilepsy included in our study reported the highest scores for the cognitive functioning and family relationships domains and the lowest scores for the worry and emotional functioning domains. This may undoubtedly be due to the concerns that the parents of children with epilepsy have about their children’s future and about how their children’s medical condition will be perceived by others. In addition, chronic conditions are often associated with a lack of prospects for normal, independent functioning. Emotional factors play a role in reducing QoL. There are reports showing that there is an association between the anxiety, stress and fear experienced by parents and the quality of their child’s life [34]. In their paper on parental concerns towards children with epilepsy, Murugupillai et al. [35] noted that these concerns are multidimensional and relate to such areas as the child’s physical, behavioural, psychological and social functioning, and education as well as treatment with anti-epileptic medications. The parents studied were also concerned that epilepsy would affect their children’s prospects for continuing education, securing a good job and getting married. Raising a child as a single parent has negative effects on the parent’s physical and social functioning and overall quality of life. In their paper, Rozenek and Owczarek [36] discussed the issue of the behaviour of the parents of children with epilepsy. The burden of a child’s illness is often such a strong stressor for parents that they are unable to cope with it on their own. It disrupts the relationship between parents and leads to conflicts and grudges, including those relating to the views on the child’s treatment as well as the restrictions and parenting methods used. When trying to deal with an excess of emotions relating to their child’s diagnosis, parents sometimes throw themselves into work or resort to alcohol. All those negative experiences are accompanied by the lack of social acceptance and support and by the constant fatigue and stress resulting from permanent alertness and an increased number of responsibilities [37]. The daily care and rehabilitation of a child with a disability as well as the disability itself are stressors conducive to the development of various disorders and burnout. This phenomenon is more likely to affect mothers than fathers, as mothers carry out a range of care, housekeeping, nursing and educational activities. Raising a child with a disability as a single parent is extremely difficult. Single parents of children with disabilities have an excessive burden of responsibilities and often have no support from people close to them. Mothers of such children are at a high risk of experiencing psychological and physical strain, and some even experience depression [38]. In the case of single parents, the everyday care of a child with an illness contributes to a reduction in social contact and the isolation of the family. In their paper on the functioning of the families of children with cerebral palsy, which, like epilepsy, is a chronic condition, Britner et al. [38] indicated that partners may be of great support to one another in dealing with a child’s illness, which confirms the results of the present study, which found that those parents and legal guardians who were in a relationship reported better family functioning. The child’s disease requires the mobilization of the family members, a mutual understanding and actions. Caring duties, together with other household work, may excessively burden the mother; therefore, this is the point when she needs the greatest support. The organisation of the home, focused on solving the problems, gives rise to a series of changes in intrafamilial relationships. This, in turn, may result in the disturbance of the emotional atmosphere in the family, due, at least in part, to the unsatisfied needs of the mother. It is pivotal to understand that a family coping with an epileptic child is an interactive process covering not only behavioural responses but also, most importantly, triggering the resources required in the process of handling a difficult situation. A child’s chronic disease is such a case for the family.

### Limitation of the Study

The findings of the present study must be interpreted with caution, with some limitations kept in mind. First, the evaluation of quality of life by a parent/legal guardians-proxy might be limited to their individual perceptions of health status and functioning among children and adolescents. The second limitation concerns the fact that quality of life was assessed on the basis of generic scales; in future studies, a child-specific questionnaire should be used.

## 5. Conclusions

Child epilepsy demonstrates a considerable impact on the quality of a child’s life and their family functions. Children with epilepsy require holistic care. Therefore, studies on the determinants of high QoL seem to be crucial for the planning of interventions aimed at maximizing QoL. Epilepsy makes social life harder, compromises relationships with peers, affects a lower self-esteem, worsens cognitive processes, and retards the achievement of self-reliance. These parameters have a considerable influence on the individual development of a child; therefore, they should be evaluated during the therapy on a constant basis. The key role should be played by social support and the provision of possibly normal functioning to the child in all the areas of its life. The treatment of epilepsy and the control of epileptic seizures are insufficient at this stage of advancements in medicine in order to provide the child with a better QoL. Therefore, care over epileptic children should focus not only on the disease signs and symptoms but also how a child and its family perceive the disease. The treatment of epilepsy is not based only on the annoying symptoms of the disease but also on the improvement of the child and its family’s QoL.

## Figures and Tables

**Table 1 ijerph-19-02277-t001:** Characteristics of the children’s parents.

Variable		Mean (SD)	Median (Quartiles)
Age of the Respondent [Years]		35.79 (5.89)	35 (32–39.5)
Variable		*N*	%
Gender	Female	96	93.20%
	Male	6	5.83%
	No answer *	1	0.97%
Education	Primary	1	0.97%
	Vocational	10	9.71%
	Secondary	46	44.66%
	Tertiary	46	44.66%
Place of residence	Urban area	67	65.05%
	Rural area	35	33.98%
	No answer *	1	0.97%
Marital status	Single	19	18.45%
	In a relationship	84	81.55%
Number of children in the family	1	36	34.95%
	2	45	43.69%
	3	17	16.50%
	4	4	3.88%
	No answer *	1	0.97%

* The respondent provided no answer.

**Table 2 ijerph-19-02277-t002:** Characteristics of the children studied.

Variable		*N*	%
Age of the child	2–4 years	25	24.27%
	5–7 years	25	24.27%
	8–12 years	35	33.98%
	13–18 years	11	10.68%
	18–25 years	6	5.83%
	>25 years	1	0.97%
Place of residence	Urban area	67	65.05%
	Rural area	35	33.98%
	No answer *	1	0.97%
Comorbidities	No	25	24.27%
	Yes	78	75.73%

* The respondent provided no answer.

**Table 3 ijerph-19-02277-t003:** Assessment of the Children’s Functioning in Particular Domains.

PedsQL Subscales	*N*	M	SD	Me	Min.	Max.	Q1	Q3
Physical functioning	103	49.45	28.43	46.88	0	100	25	76.56
Emotional functioning	103	43.4	21.39	45	0	100	30	55
Social functioning	103	49.47	27.39	45	0	100	25	67.5
Work/school functioning	103	42.36	27.89	40	0	100	25	55
Psychosocial health	103	44.9	20.39	41.67	1.67	98.08	31.67	55.83
Total QoL score	103	46.42	20.95	46.43	1.09	92.86	31.52	60.33

M, mean; Me, median; Min., minimum; Max., maximum; SD, standard deviation.

**Table 4 ijerph-19-02277-t004:** Quality of life of children with epilepsy in relation to their age.

PedsQL	Age of the Child	*N*	M	SD	Me	Min.	Max.	*p* *
Physical functioning	2–4 years (A)	25	49.38	31.3	43.75	0	93.75	0.832
	5–7 years (B)	25	44.38	27.83	46.88	0	87.5	
	8–12 years (C)	35	52.14	23.37	50	0	100	
	>12 years (D)	18	51.39	34.97	70.31	0	96.88	
Emotional functioning	2–4 years (A)	25	50	25.33	50	0	100	0.326
	5–7 years (B)	25	45	19.79	45	5	85	
	8–12 years (C)	35	39	19.01	40	0	80	
	>12 years (D)	18	40.56	21.14	40	0	70	
Social functioning	2–4 years (A)	25	60.6	30.05	50	15	100	0.193
	5–7 years (B)	25	46.6	29.46	45	0	100	
	8–12 years (C)	35	42.86	20.23	45	0	85	
	>12 years (D)	18	50.83	29.91	52.5	0	90	
Work/school functioning	2–4 years (A)	25	62.33	34.62	50	0	100	0.012
	5–7 years (B)	25	35.2	28.85	25	0	85	A>
	8–12 years (C)	35	36.43	15.17	35	5	70	B, C
	>12 years (D)	18	36.11	23.74	37.5	0	70	
Psychosocial health	2–4 years (A)	25	56.92	24.86	50	7.69	98.08	0.047
	5–7 years (B)	25	42.27	23.35	38.33	6.67	83.33	A > C
	8–12 years (C)	35	39.43	13.34	40	1.67	66.67	
	>12 years (D)	18	42.5	14.36	40	21.67	68.33	
Total QoL score	2–4 years (A)	25	54.05	26.56	47.62	4.76	92.86	0.412
	5–7 years (B)	25	43	23.17	44.57	6.52	83.7	
	8–12 years (C)	35	43.85	14.23	42.39	1.09	68.48	
	>12 years (D)	18	45.59	18.89	50	14.13	72.83	

* Kruskal-Wallis test + post-hoc analysis (Dunn’s test); M, mean; Me, median; Min., minimum; Max., maximum; SD, standard deviation.

**Table 5 ijerph-19-02277-t005:** Relationship between QoL of the children and the parental age.

PedsQL	Correlation with the Age of the Respondent			
	Correlation Coefficient	*p*	Direction of Relationship	Strength of Relationship
Physical functioning	0.001	0.991	-	-
Emotional functioning	0.065	0.516	-	-
Social functioning	−0.071	0.478	-	-
Work/school functioning	−0.204	0.038	Negative	very weak
Psychosocial health	−0.134	0.178	-	-
Total QoL score	−0.077	0.44	-	-

**Table 6 ijerph-19-02277-t006:** Assessment of the parents’ functioning in individual domains.

PedsQL—Family Impact Module Subscales	*N*	M	SD	Me	Min.	Max.	Q1	Q3
Physical functioning	103	41.02	20.44	41.67	0	100	27.08	54.17
Emotional functioning	103	39.66	20.59	40	0	95	25	50
Social functioning	103	43.08	25.36	43.75	0	100	25	62.5
Cognitive functioning	103	52.86	27.98	55	0	100	30	75
Communication	103	43.45	23.44	41.67	0	100	25	58.33
Worry	103	24.17	19.5	25	0	100	7.5	35
Daily activities	103	32.44	26.5	25	0	100	8.33	50
Family relationships	103	55.63	24.03	50	10	100	35	75

M, mean; Me, median; Min., minimum; Max., maximum; SD, standard deviation.

**Table 7 ijerph-19-02277-t007:** Summary scores.

PedsQL—Family Impact Module Subscales	*N*	M	SD	Me	Min.	Max.	Q1	Q3
Parent QoL summary score	103	44.05	18.93	42.5	1.25	98.75	30.62	55
Family functioning summary score	103	46.94	21.94	43.75	9.38	100	31.25	64.06
Total impact score	103	41.88	17.35	40.28	4.86	94.44	30.21	51.04

M, mean; Me, median; Min., minimum; Max., maximum; SD, standard deviation.

**Table 8 ijerph-19-02277-t008:** Relationship between QoL in the children with epilepsy and the presence of comorbidities.

PedsQL	Comorbidities	*N*	M	SD	Me	Min.	Max.	*p*
Physical functioning	No	25	58.88	25.49	62.5	0	93.75	0.047
	Yes	78	46.43	28.81	43.75	0	100	
Emotional functioning	No	25	50.8	21.34	50	0	100	0.07
	Yes	78	41.03	20.99	40	0	85	
Social functioning	No	25	63.4	24.18	65	0	100	0.002
	Yes	78	45	26.98	45	0	100	
Work/school functioning	No	25	53.53	27.29	50	10	100	0.022
	Yes	78	38.78	27.29	35	0	100	
Psychosocial health	No	25	55.59	19.45	50	13.33	98.08	0.002
	Yes	78	41.47	19.6	38.33	1.67	90.38	
Total QoL score	No	25	56.74	19.5	55.43	19.57	92.86	0.005
	Yes	78	43.12	20.42	41.85	1.09	91.67	

M, mean; Me, median; Min., minimum; Max., maximum; SD, standard deviation.

**Table 9 ijerph-19-02277-t009:** Independent predictors of QoL in the PedsQL questionnaire affecting total QoL scores.

Variable		Regression Parameter	95% CI		*p*
Age of the respondent [years]		0.142	−0.597	0.882	0.702
PedsQL—Family Impact Module	Physical functioning	0.108	−0.174	0.391	0.447
	Emotional functioning	−0.123	−0.39	0.144	0.363
	Social functioning	0.257	0.057	0.457	0.013
	Cognitive functioning	0.232	0.061	0.403	0.009
	Communication	−0.126	−0.351	0.098	0.266
	Worry	0.141	−0.108	0.39	0.265
	Daily activities	0.144	−0.077	0.366	0.199
	Family relationships	−0.069	−0.29	0.152	0.536
Age of the child	2–4 years	ref.			
	5–7 years	−11.956	−23.022	−0.89	0.035
	8–12 years	−10.829	−21.947	0.289	0.056
	>12 years	−7.064	−20.893	6.765	0.312
Gender	Female	ref.			
	Male	−10.898	−26.935	5.14	0.18
Education	Primary, vocational	ref.			
	Secondary	−2.777	−14.696	9.141	0.644
	Tertiary	3.521	−9.211	16.253	0.584
Place of residence	Urban area	ref.			
	Rural area	6.332	−1.976	14.641	0.133
Marital status	Single	ref.			
	In a relationship	−4.968	−14.554	4.617	0.305
Number of children in the family	1	ref.			
	2	7.82	−0.444	16.084	0.063
	3 or 4	15.957	5.238	26.676	0.004
Comorbidities	No	ref.			
	Yes	−6.573	−16.342	3.196	0.184

## Data Availability

The data presented in this study are available on request from the corresponding author.

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
