# Peer review of "The Quality of Life of Children with Epilepsy and the Impact of the Disease on the Family Functioning"

_ijerph, 2022, doi:10.3390/ijerph19042277_

Round 1
Reviewer 1 Report
The manuscript “The Quality of Life of Children with Epilepsy and the Impact of the Disease on the Family Functioning” describes cross-sectional study of a sample of 103 epilepsy patients’ caregivers, who provided information about quality of life of their children and evaluated impact of the disease on the family functioning. QoL studies are important when trying to understand burden of the disease as well as looking for the resilience resources that can be used in psychosocial interventions. However, I would like to draw attention to several aspects of the current manuscript that might be improved or needs to be clarified.
- Abstract (lines 15-18). “The parents reported the highest scores…” While reading an abstract reader is still not introduced to the measures that were used in the study and have no information about direction of interpretation of provided scores, e.g. whether high scores indicate good or decreased functioning and QoL. I would suggest changing formulation of the results in the abstract and providing reader with interpretation e.g. children with epilepsy demonstrated decreased emotional functioning”.
- Please provide information regarding language version of instruments that were used in current study. What language was used for the assessment? If it was Polish, please provide information about Polish versions of QoL instruments– translation, adaptation, previous research on psychometric properties of these questionnaires.
- Please provide rationale why epilepsy patients older that 18 years were considered as children. Was PedsQL 4.0 sensitive enough to reflect quality of functioning of older subjects?
- While describing a sample of epilepsy patients please provide clinical characteristic, including number of subjects with developmental disorders, mental retardation and special needs, frequency of seizures and other information that would help to evaluate actual medical burden of disease.
- Table 9 might be omitted. If authors think that this table is needed, it could be added as a supplementary table. Text explanations of the direction and strength of the correlations are not necessary, because this information is obvious from provided correlation scores. Simple correlation matrix is a standard for presenting results of correlation analysis.
- Linear regression analysis – please describe all the variables that were included into analysis and rationale why they were chosen. Variables that are neither scale nor dichotomous are generally not recommended for linear regression. Please provide explanation why they were not recoded. Please also included data on multicollinearity of variables.
- Generally, more clearly described logic behind the study would help to follow results easier. Please include step by step explanation of your analysis and how that contribute to the main objective of current research.
- Concluding part of the discussion might be slightly more elaborated avoiding very general statements such as “It is necessary to conduct further studies on QoL in children with epilepsy and their parents or careers”. Please provide more specific recommendation.
Author Response
Thank you very much for sending us the consensus opinion about requested revision of our manuscript entitled: The Quality of Life of Children with Epilepsy and the Impact of the Disease on the Family Functioning. The manuscript had been revised according to the reviewers comments and criticism. Most suggestions were accepted and incorporated into the text.

Reviewer 2 Report
This work examined the Quality of Life of children with epilepsy and the impact of the disease on family functioning. The work considers, as a source of data, the collection of information through the application of standardized questionnaires (PedsQL 4.0 and PedsQL 2.0) and its own questionnaire. The study included children of different ages, concentrating on children between 2 and 7 years old; most urban areas and with comorbidities. This is very appropriate, given that this disease occurs with a higher rate at these ages. The work stands out for having carried out the study in a real way, for the use of widely recognized data collection instruments and for using different statistical methods for the quantitative study. The authors emphasize that QoL in children with epilepsy was highest in the social and physical domains and lowest in the preschool/school/work functioning domain. In addition, it is very evident that the older children had a lower overall QoL, as measured by the total QoL score, compared with younger children. Regarding the parents, the results show a great concern for the future of their children, in the context of social acceptance, their work-life and functional independence. The work offers an extremely important topic and applies a good methodology; however, it can improve substantially if the following recommendations are followed:
In the Summary section, we must highlight the contribution of the work to the subject, compared with previous works, to make clear the importance of its study.
In the Introduction section, the authors must further describe the problem and the objective of the work (the quality of life of children with epilepsy and the impact of the disease on family functioning). Since most of the text, in this section, is focused on the state of epilepsy in the world and the advantages of early detection, but they do not discuss the problems and usefulness of their work.
In the Results section, although the tables have excellent information and are well detailed, the explanation is not, which is very short. They should highlight the most important findings. For example, in Table 2, why is the percentage of women considered in the study so high; why most children are from urban areas. In section 3.1.1., the QoL value of 46, but how good, within this context of life. Table 5 is not very clear in its description or explanation.
The Discussion Section is generally good. However, the authors must not abuse the citation of other works; they should concentrate more on their own discussion of the findings, both the results and the process of applying the questionnaires.
In the Conclusions section, the authors could add the contributions of their work and the most significant results.
Author Response

(The authors gave the same response as above.)

Reviewer 3 Report
The authors present a cross-sectional study measuring the quality of life (QoL) in children with epilepsy. The methods are clearly described, and the design is correct. It shows interesting results that merit its publication.
The are some concerns to be solved:
- In the inclusion criteria, the authors have collected whether the patients had or not comorbidities. However, this is poorly specified and quite variable. I should recommend describing which comorbidities were included o searched for, in more detail.
- It should also be useful to know what kind of epilepsy the patients had (eg. Tonic-clonic generalized seizures, non-convulsive, etc). And whether there were differences in QoL among them.
- Some English editing should be revised. For example, the term “careers” should be changed to “carers” all throughout the manuscript.
Author Response

(The authors gave the same response as above.)
